# Relationships between Seminal Plasma Metabolites, Semen Characteristics and Sperm Kinetics in Donkey (*Equus asinus*)

**DOI:** 10.3390/ani11010201

**Published:** 2021-01-15

**Authors:** Maria Antonietta Castiglione Morelli, Angela Ostuni, Brunella Giangaspero, Stefano Cecchini, Augusto Carluccio, Raffaele Boni

**Affiliations:** 1Department of Sciences, Campus Macchia Romana, University of Basilicata, 85100 Potenza, Italy; maria.castiglione@unibas.it (M.A.C.M.); angela.ostuni@unibas.it (A.O.); stefano.cecchini@unibas.it (S.C.); 2Faculty of Veterinary Medicine, University of Teramo, Loc. Piano d’Accio, 64100 Teramo, Italy; brunellagiangaspero@gmail.com

**Keywords:** donkey, seminal plasma, metabolites, ^1^H NMR, sperm kinetics

## Abstract

**Simple Summary:**

A deeper knowledge of reproductive biology may be helpful in the donkey to avoid the risk of extinction that some breeds are facing. The evaluation of metabolites in seminal plasma provides crucial information for the knowledge of donkey sperm metabolism, for obtaining comparative information with other species, as well as for providing useful elements for the formulation of extenders for sperm dilution and conservation. Moreover, correlations of seminal metabolites with sperm kinetics highlight new possible markers of sperm quality. Using multivariate analysis, all metabolic, seminal, and spermatic data were merged in a single dot that grouped individual stallions within clusters in the Cartesian axes according to the different spermatic characteristics. This amount of information also allows to shed light on the effects of total or partial removal of seminal plasma for improving sperm preservation. The inclusion in the study of an azoospermic individual represents a further discriminating element in the analysis of sperm quality under physiological and pathological conditions.

**Abstract:**

This study aimed to evaluate donkey seminal plasma metabolites and relate this information to the main characteristics of sperm quality. Sperm kinetics from 10 donkey stallions were analyzed with a computerized system at the time of collection (T0) and after 24 h storage at 4 °C (T24). Seminal plasma was frozen at −80 °C for subsequent proton nuclear magnetic resonance (^1^H NMR) spectroscopy. On three stallions, semen collection was repeated monthly for three times and sperm analysis also included mitochondrial activity and oxidative status. One stallion was azoospermic and a second semen collection was performed after one month. In the seminal plasma, 17 metabolites were identified; their levels showed numerous significant variations between the azoospermic and the normospermic individuals and grouped in well-defined clusters in a multivariate analysis. Comparing individuals with high and low sperm motility, the only discriminating metabolite was phenylalanine, whose levels were lower in the latter, as in the azoospermic individual. Phenylalanine was also the only metabolite highly correlated with all sperm kinematic parameters at T24. In conclusion, the present study has provided relevant information on the chemical characteristics of donkey semen, identified relationships between seminal metabolites, semen parameters, and sperm kinetics, and offered insights for future technological applications.

## 1. Introduction

Seminal plasma (SP) is the liquid component of the semen that originates from the intratubular liquid of the testicle and epididymis and is enriched by the secretion of the accessory sexual glands (prostate, vesicular and bulbourethral glands). These constituents join in a mix or are mostly emitted in different jets during ejaculation, contributing to the formation of spermatic fractions, which play peculiar roles in reproductive mechanisms. In donkey (*Equus asinus*), the information on the characteristics of SP is very scarce. In the *Equus* species, a multi-jet ejaculation and the production of an abundant ejaculate volume contribute to the possibility of differentiating the various spermatic fractions. However, in the practice of semen collection, all the seminal material is collected in a single mixture, which is usually filtered to remove the gelatinous part that normally constitutes the last spermatic fraction, hardly miscible with the others and with the extender. In horses (*Equus caballus*), 6–9 spermatic fractions have been described during an ejaculation [1]. The first portion is attributable to the bulbourethral glands, then, there is the contribution of the ampulla of ductus deferens, epididymal and prostate fluids while the terminal fraction is mainly released by the vesicular glands [2,3].

Although in the natural mating the semen passes directly from the male to the female urogenital tract, where spermatozoa quickly separate from the SP reaching the oviduct, the SP plays an important role not only in maintaining a state of fitness of the sperm population but also in interacting with the uterine mucosa to ensure reproduction. However, SP would not always play a beneficial role on the sperm population and the endometrium. In support of this statement, there are numerous studies that have evaluated how the removal of SP by centrifugation improves the efficiency of artificial insemination (AI) or sperm cryopreservation both in the asinine [4,5,6] and in the equine [7] species. However, the presence of SP seems to decrease the uterine inflammatory response following AI [8]. In fact, by in vitro co-incubating donkey sperm with uterine Jennie’s secretions, a lower number of sperm–polymorphonuclear neutrophils attachment was found rather than in the absence of SP.

Due to its peculiar characteristics and properties, SP may provide valuable diagnostic and predictive information for assessing the sperm quality and, consequently, estimating the male fertility. Along this line, numerous studies have been carried out on this matter in humans [9] and other animal species [10,11], where AI is a consolidated technique.

Proton nuclear magnetic resonance (^1^H NMR) applies nuclear magnetic resonance spectroscopy to the hydrogen-1 nucleus within a molecule in order to derive its molecular structure [12]. This technology allows to determine small molecule substrates, intermediates and end products of cell/tissue metabolism with simple spectra and to provide an instantaneous snapshot of the cell/tissue metabolic activity [13]. Analyses of SP by ^1^H NMR have been already performed in human [14,15], bovine [16], and equine [17] species. These studies have provided interesting associations between some metabolites and parameters related to sperm function and generated a considerable amount of information that could be very useful for a better knowledge of sperm physiology as well as for the development of synthetic semen extenders.

The present study was aimed to carry out a first exploration on the presence and concentration of metabolites in donkey SP, correlating them with parameters related to sperm function. In some stallions, the sperm collection and SP analysis were repeated in the following two months for evaluating the metabolite variability within the same individual and expanding the analysis spectrum to sperm mitochondrial activity and oxidative status. Enrolling a stallion affected by azoospermia provided a test stud for comparing physiological and pathological conditions. Information on sperm quality plays an important role in the donkey considering the population reduction that this species has experienced, putting the survival of some of its breeds at risk.

## 2. Materials and Methods

### 2.1. Reagents

Polyvinyl alcohol (PVA), potassium hydroxide (KOH), CuSO_4_, dimethyl sulfoxide (DMSO), menadione and 5,5′,6,6′-tetrachloro-1,1′,3,3′-tetraethylbenzimidazolyl-carbocyanine iodide (JC-1), deuterated water, 3-trimethylsilyl propionic acid-d4 sodium salt (TSP), and water were purchased from Sigma Chemical Company (Milan, Italy) and cell culture tested. Then, 4,4-difluoro-5-(4-phenyl-1,3, butadienyl) 4-bora-3a,4a-diaza-s-indacene-33-undecanoic acid (C11-BODIPY^581/591^) was obtained from Life Technologies (Milan, Italy). Hydrogen peroxide solution 30% was purchased by Biochem (Chemopharma, Cosne sur Loire, France). Phosphate buffer saline (PBS, pH 7.4, cell culture tested) was purchased from Gibco (Life technologies, Grand Island, NY, USA).

### 2.2. Animals

Nine adult and reproductively mature Martina Franca (MF) jackasses, aging from 2 to 18 years and weighing from 150 to 450 kg, and one adult and reproductively mature Romanian (RO) jackass, 11 years old and weighing 185 kg, were enrolled in this study. All stallions were trained for semen collection and used in AI programs with the exception of one of the nine MF stallions suffering from a condition of azoospermia. Seven MF jackasses (six normospermic and one azoospermic individuals) were housed at the Rustic Found of Chiareto, Veterinary Faculty, University of Teramo (Italy). The remaining three jackasses (2 MF and 1 RO) were housed at a private farm in the Potenza district (Italy). In both farms, jackasses were kept in box stalls with an open paddock, under natural light conditions. 

### 2.3. Semen Collection

Semen was collected from March to June. Prior to beginning of the experiment (up to 3–4 days), jackasses were submitted at least three preliminary semen collections scheduled twice weekly. By using a Missouri artificial vagina, semen was collected and, then, in the laboratory, filtered through a sterile gauze. Semen volume was evaluated by a measuring cylinder whereas sperm concentration was estimated by a Makler camera (Sefi-Medical Instrument, Haifa, Israel). Sperm production was obtained by sperm concentration x semen volume. A homogeneous part of the semen was centrifuged (2000× *g* for 10 min) to separate the SP from cells and debris. Another part was diluted with INRA96 (IMV Technologies, L’Aigle, France) (1:3-volume of semen: volume of extender), stored within a syringe (Becton, Dickinson and Co, Franklin Lakes, NJ, USA) to obtain anaerobic conditions and slowly cooled at +4 °C. Sperm kinetics was again evaluated in the refrigerated sperm 24 h after collection.

### 2.4. Experimental Design

In Experiment 1, seven MF stallions housed in Teramo district were submitted to semen collection. A single ejaculate was used and semen characteristics as well as sperm kinetics at either the time of collection (T0) or 24 h later (T24) were analyzed. One stallion was azoospermic and the semen collection was repeated after one month. The remaining six stallions were selected on the basis of their total sperm motility (Tot Mot) and grouped in a high motility group (Tot Mot > 70%, *n* = 3) and a low motility group (Tot Mot < 70%, *n* = 3). To support this classification based on in vitro analysis, the fertility data of these stallions were collected during the breeding season of this study. Artificial insemination (AI) was carried out using doses of fresh semen containing an equal number of motile spermatozoa on natural heats and human chorionic gonadotropin (hCG)-induced ovulations [18].

In Experiment 2, three stallions housed in Potenza district (2 MF and 1 RO jackasses) were submitted to repeated semen collections on a monthly basis for three times. Together with semen characteristics and sperm kinetics, sperm analysis considered the bioenergetic and oxidative status of the spermatozoa at either the time of sperm collection (T0) or 24 h later (T24).

The SP of the ten donkey stallions was analyzed by ^1^H-NMR spectroscopy. The levels of metabolites detected were related to semen characteristics, sperm kinetics, as well as to sperm bioenergetic and oxidative status in the Experiment 2.

### 2.5. Sperm Kinetics

Sperm kinetics were evaluated with either SCA 5.0 (Microptic, Barcelona, Spain) [19] or CASA IVOS 12.3 (Hamilton Thorne Biosciences, Beverly, MA, USA) [20] systems. Before the beginning of the experiment, both systems were compared by cross-checking with frozen semen and properly calibrated. Sperm concentration was adjusted with INRA96 to 20–30 × 10^6^ spermatozoa/mL and samples were equilibrated for 2 min at 37 °C. Spermatozoa with an average velocity of less than 10 µm/s were considered immotile. Sperm kinetics included: the percentage of motile spermatozoa (Tot Mot); the percentage of progressive spermatozoa (Prog, average path velocity higher than 70 µm/s and straightness of track higher than 80%); the curvilinear velocity (VCL, µm/s); the straight-line velocity (VSL, µm/s); and the average path velocity (VAP, µm/s).

### 2.6. Mitochondrial Membrane Potential (MMP)

JC-1 is a dye exhibiting potential-dependent accumulation in mitochondria [19]. MMP was evaluated as previously described [21]. In brief, spermatozoa (1 × 10^6^) were incubated with 1.5 µM JC-1 in PBS supplemented with 0.5% PVA (PBS-PVA) at 37 °C for 30 min, washed by centrifugation and again incubated for 30 min at 37 °C. Samples were read with a spectrofluorometer (Cary Eclipse, Agilent Technologies, Rome, Italy). MMP was measured as the ratio between the fluorescence intensity peak values at ~595 nm (FoB) and ~535 nm (FoA).

### 2.7. Lipid Peroxidation (LPO)

LPO was determined by measuring C11-BODIPY^581/591^ fluorescence, as previously described [21]. Sperm aliquots (1 × 10^6^ spermatozoa/mL) were incubated with 2 µM C11-BODIPY^581/591^ for 30 min at 37 °C, centrifuged at 230× *g* for 10 min, incubated in PBS-PVA at 37 °C for 30 min, and read with a spectrofluorometer. LPO was evaluated as the ratio between the Fo values at ~515 nm (FoA) and the sum of the fluorescence peak values at ~515 (FoA) and ~590 nm (FoB).

### 2.8. Anti-LPO Potential

Anti-LPO potential evaluated the ability of sperm defenses to counteract oxidative stress [21]. In brief, spermatozoa (1 × 10^6^) were incubated in the dark for 30 min at 37 °C with C11-BODIPY^581/591^ and 0.5% OxMix, an oxidant solution containing 2 mM menadione, 1.8 mM CuSO_4_, and 0.25 mM H_2_O_2_ [22]. Spermatozoa were then washed by centrifugation at 230× g for 10 min and incubated further for 30 min at 37 °C in PBS-PVA. Samples were read with a spectrofluorometer, as above. Anti-LPO potential was evaluated as the ratio between the fluorescence intensity peak values at ~590 nm (FoB) and the sum of the fluorescence peak values at ~515 (FoA) and ~590 nm (FoB).

### 2.9. Nitroblue Tetrazolium (NBT) Assay

NBT assay evaluates the production of superoxide anions in cells [21,23]. Sperm suspensions (15 × 10^6^ spermatozoa) were centrifuged at 230× *g* for 5 min, the pellets were incubated with NBT solution for 45 min at 38 °C [23], then washed with PBS-PVA. Intracellular formazan was solubilized with 2 M KOH: DMSO solution and read with a spectrophotometer (Bio-Rad 550, Hercules, CA, USA) at 630 nm. Superoxide anion production (µg formazan) was calculated by using a standard curve of absorbance values.

### 2.10. ^1^H NMR Sample Preparation

Thawed SP samples were microfiltered (Vivaspin 20; 3000 MWCO, Sartorius, Göttingen, Germany) to remove high molecular weight components by centrifugation at 13,000× *g* for 30 min and kept at 4 °C before being processed within a few hours.

### 2.11. ^1^H NMR Analysis

An amount of 200 µL of SP was mixed with 300 µL of deuterium oxide and 5 μL of TSP (3-trimethylsilyl propionic acid-d4 sodium salt). TSP was used as both chemical shift reference (δ = 0) and internal standard for quantitative analysis.

^1^H NMR spectra were acquired at 298 K on a Varian Unity Inova spectrometer (Varian, Palo Alto, CA, USA) operating at 500 MHz. ^1^H NMR spectra of the solutions were acquired using presaturation of the solvent signal. 256 free induction decays (FIDs) were collected into 32 K data points using a spectral width of 12 ppm, with a 7.6 µs pulse width and a relaxation delay of 2 s, with an acquisition time of 3 s. VNMRJ 2.1B software (Agilent Technologies, Santa Clara, CA, USA) was used to acquire all the spectra that were processed using NMR SUITE 8.0 (Chenomx Inc., Edmonton, Alberta, Canada). Then, 1D spectra were Fourier transformed with FT size of 132 k and a 1 Hz line-broadening, phased, and polynomial baseline corrections were applied over the whole spectral range. The PROFILER module was used to identify and quantify metabolites by fitting the compound signatures from the spectral Chenomx NMR Suite library with patterns in sample spectra.

### 2.12. Statistical Analysis

In experiment 1, data were obtained from six normospermic stallions and one stallion with azoospermia. In experiment 2, data were collected from three normospermic stallions in which sperm collection was repeated three times on a monthly basis (3 ejaculates × 3 jackasses). The dataset was organized, as follows: the experimental date, the replication number, the 17 SP metabolites, five semen characteristics (i.e., semen pH, osmolarity and volume as well as sperm concentration and production) and nine sperm quality endpoints (i.e., total and progressive motility, VCL, VSL, VAP, MMP, LPO, anti-LPO potential and NBT values).

The Shapiro-Wilk test was applied to evaluate the normal data distribution and the Levene’s test was used to evaluate the homogeneity assumption needed for carrying out parametric tests. Variables measured in percentages were transformed into angles corresponding to arcsine of the square root for variance analyses. The pH values did not follow a continuous distribution, so H^+^ concentrations were log transformed before the analysis. Tabular data were analyzed by ANOVA (Systat 11.0, Systat Software, Inc., San Jose, CA, USA); they are presented as non-transformed values, for ease of interpretation. Pair-wise comparisons of the means were performed with Fisher’s least significant differences (LSD) test. For Experiment 2, data were first analyzed by using the coefficient of variation (CV)-ANOVA [24] and then by repeated measures ANOVA (Systat 11.0). Coefficients of correlation (R) were calculated by linear regression procedure (Systat 11.0). The minimum level of statistical significance was *p* < 0.05. Values are presented as mean ± standard deviation (SD).

Multivariate analyses were performed using different data sets: in a first analysis, only NMR data were used. In the second analysis, both NMR and data associated with semen and sperm quality were employed. Data were imported into the SIMPCA-P+ software (Version 12, Umetrics, Sweden). Analyses were performed in two steps: a preliminary multivariate statistical analysis was performed with the principal component analysis (PCA), and in the second step, partial least squares discriminant analysis (PLS-DA) was carried out. Data for PCA were subjected to pre-treatment with Pareto scaling (/√SD) which automatically mean-centers the data; while data for PLS-DA were UV scaled.

PCA score and loading plots were interpreted to find out whether a separation between different samples was possible. Similarly, PLS-DA provided a visual interpretation of the data through the two-dimensional score plot that showed the separation between different classes. Furthermore, PLS-DA gave other statistics such as the variable importance in projection (VIP) that highlights the importance of each variable in projection, and R^2^ and Q^2^ statistics, which were used to evaluate the predictive ability of the PLS-DA model, in which R^2^ measures the goodness of fit and displays the explained variation by components and Q^2^ gives an indication of the goodness of predicted model [25]. PLS-DA models were validated by using permutation tests.

## 3. Results

From ^1^H NMR spectra of the SP, 17 metabolites were identified (Table 1 and Table 2). These metabolites can be grouped into: amino acids, as alanine, glutamate, leucine, and phenylalanine or amino acid derivative, as creatine; metabolic compounds, as acetate, o-acetylcarnitine (ALC), carnitine, choline, citrate, glycerophosphocholine (GPC), lactate, myo-inositol, and trimethylamine N-oxide (TMAO); intermediate metabolites of the degradation of phenols, as hippurate; substances with antimicrobial activity, as benzoate; or components with unknown activities, as glycerol.

### 3.1. Experiment 1. Azoospermic vs. Normospermic Stallions

The concentrations of SP metabolites between the azoospermic and the normospermic individuals showed a greater number (*p* < 0.05) of differences compared to that between the individuals with high and low sperm motility (Table 1). In particular, the azoospermic stallion showed lower (*p* < 0.05) levels of alanine, carnitine, choline, citrate, creatine, glutamate, lactate, leucine, myoinositol, and GPC compared to the normospermic stallions. The semen volume was significantly (*p* < 0.05) lower whereas the pH values were significantly (*p* < 0.01) higher in the azoo- than in the normospermic stallions. Regarding the comparison between both groups of normospermic subjects, only phenylamine significantly (*p* < 0.05) differed between groups, showing a lower content in the lower motility group, and approaching that recorded in the azoospermic stallion (0.009 ± 0.005 vs. 0.001 ± 0.002 vs. 0.000 ± 0.000, *p* < 0.05, respectively). As far as sperm kinetics is concerned, at T0, a statistically significant difference between the two groups was found about Tot Mot (93.1 ± 6.1 vs. 56.3 ± 17.0%; *p* < 0.05) and Prog (33.9 ± 6.0 vs. 17.7 ± 3.2%; *p* < 0.01). This discrepancy was significantly accentuated at 24 h also involving the other sperm kinematic parameters as VCL, VSL and VAP. For the production of mules, a total number of 46 mares were artificially inseminated with high (*n* = 3) and low (*n* = 3) sperm motility donkey stallions involved in this study (17 and 29 mares, respectively). No significant differences emerged between these two groups regarding fertility rate (number of pregnancies/number of inseminated females), number of cycles inseminated/pregnancy, and number of inseminations/pregnancy (88.23 vs. 82.75%, 1.33 vs. 1.42, 1.87 vs. 2.13, respectively).

### 3.2. Experiment 2. Repeated Semen Collections

Analysis by CV-ANOVA of the 17 metabolites detected in the SP of the three stallions subjected to three repeated sperm collection on a monthly basis showed variability between individuals significantly (*p* < 0.01) higher than that recorded within individuals. The average results obtained in Experiment 2 are reported in Table 2. RO stallion showed a lower sperm concentration than MF1 stallion but higher pH, osmolarity, and lactate values than MF1 and MF2 stallions. The SP glutamate levels were lower in RO and MF2 stallions than MF1 stallion whereas the hippurate levels in RO and MF1 stallions were lower than in MF2 stallion. Significant (*p* < 0.05) differences between the sperm kinematic characteristics at T0 were found between MF1 and MF2 stallions about VCL (110.0 ± 7.1 vs. 127.7 ± 7.9 µm/s; *p* < 0.05) and VAP (77.8 ± 10.5 vs. 95.9 ± 6.7 µm/s; *p* < 0.05). At T24, Tot Mot and Prog significantly differed between MF1 and RO stallions (75.5 ± 5.3 vs. 40.5 ± 20.9%; *p* < 0.05 and 33.5 ± 2.5 vs. 10.4 ± 10.7%; *p* < 0.01, respectively). The evaluation of mitochondrial activity as well as the examination of oxidative status did not significantly differ between stallions.

An analysis aimed at evaluating the correlations between metabolites found in SP is reported in Table 3 and Appendix A. Numerous significant (*p* < 0.05) relationships emerged between alanine, ALC, carnitine, citrate, choline, creatine, glutamate, GPC, leucine, myo-inositol, and TMAO. Acetate positively correlated (*p* < 0.05) with citrate and glutamate. Lactate showed a significant (*p* < 0.05) relationship only with citrate while the remaining SP metabolites, i.e., benzoate, glycerol, hippurate, and phenylalanine, were found to be independent of the other metabolites.

We also tested possible correlations between SP metabolites and seminal parameters, such as osmolarity, volume, sperm concentration and sperm production. No significant (*p* > 0.05) correlations emerged between osmolarity and the SP metabolites (data not shown). Semen volume significantly (*p* < 0.05) correlated with benzoate while numerous significant correlations emerged between sperm concentration and production and the SP metabolites (Table 4).

Significant (*p* < 0.05) correlations were also found between SP metabolites and sperm kinematic parameters evaluated at T0 and T24 (Table 5 and Appendix A). Aside from significant correlations showing a spot distribution, phenylalanine exhibited significant (*p* < 0.05) correlations with all kinematic parameters assessed at T24. Furthermore, at T24, all the kinematic parameters correlated with each other, whereas a lower relationship among these parameters was recorded at T0.

### 3.3. Multivariate Analysis

Multivariate statistical analyses were conducted with PCA and PLS-DA in two ways, using: (i) only NMR data (Figure 1); (ii) NMR data and seminal and sperm quality parameters (Figure 2). On the basis of NMR data, the separation of different animals from the PCA score plot was not very clear (Figure 1A). The first component of this PCA model accounted for 58.56% of total variability and the second component accounted for 23.58% of total variability, which results in a cumulative R^2^ value of 0.822. The corresponding PLS-DA model had a R^2^X value of 0.567, a R^2^Y value of 0.652, and a Q^2^ value of 0.18 (Figure 1B). A better separation of the animals using the same data set was achieved in this case. The analysis of the importance of NMR variables in discrimination of the animals (VIP parameters) is reported in Figure 1C, and the variables with VIP values > 1.0 were chosen as major discriminating metabolites. Based on this criterion, stallions could be discriminated on carnitine, phenylalanine, acetate, citrate, choline, lactate, leucine, and TMAO.

Likewise, using ^1^H NMR, semen and sperm quality data, the separation of different animals from the PCA score plot was not evident (Figure 2A). The corresponding PLS-DA model (Figure 2B) had a R^2^X value of 0.49, a R^2^Y value of 0.737, and a Q^2^ value of 0.265 and showed a better separation of the animals. The analysis of all VIP parameters is reported in Figure 2C. The discriminating variables for this model with VIP > 1.0 were: Tot Mot, Prog, osmolarity, carnitine, citrate, VSL, choline, acetate, leucine, and phenylalanine.

## 4. Discussion

The NMR analysis allowed the identification and the quantification of 17 metabolites in donkey SP. Most of these metabolites varied significantly between the azoospermic and the normospermic stallions. However, only phenylalanine significantly discriminated the SP of stallions with higher and lower sperm motility rate. To our knowledge, this is the first study that evaluated the potential relationship between SP metabolites, semen characteristics and sperm kinetics in the donkey. Magistrini et al. [17] used ^1^H NMR spectroscopy to evaluate the horse SP metabolites tracking their origin within the genital tract. Comparing donkey and horse SP metabolites, lower concentrations of alanine and carnitine were found in donkeys and, although these values fall within the range of those observed in the horse, they are placed in the lower limit of this range. On the contrary, in donkey, creatine levels lay in the upper limit of the horse creatine level range. However, no relevant differences emerged about acetate, glutamate, citrate, and lactate between equine and donkey SP.

The analysis of the SP metabolites has been evaluated in many studies in animals [16,17] and humans [14,15]. We aimed to obtain useful information on male fertility as well as for the development of extenders able to prolong the extracorporeal conservation of the semen. In donkey, some of these SP metabolites are not associated with others, as benzoate, glycerol, hippurate, and phenylalanine. Their VIP scores were lower than one. Furthermore, the levels of acetate, benzoate glycerol, and hippurate do not show significant variations between azoo- and normo-spermic individuals, between stallions with higher and lower sperm motility, as well as about seminal characteristics and sperm kinetics. This makes these SP metabolites of little interest for their potential use in discriminating elements in the evaluation of sperm quality.

In donkey SP, lactate levels were significantly (*p* < 0.05) lower in azoo- than in normo-spermic individuals and in MF than RO stallions (*p* < 0.01). We are unable to explain the latter finding that may be race related. Lactate together with pyruvate is commonly used as energy substrates by mammalian spermatozoa. Cytosolic lactate can also be transported inside mitochondria of sperm cells by the mitochondrial lactate carrier for further metabolization [26]. Phenylalanine was the most interesting SP metabolite whose levels were lower in the azoospermic and low sperm motility stallions than in high sperm motility stallions and highly correlated with all sperm kinematic parameters at T24. Phenylalanine is an essential aromatic amino acid that, when added to a culture medium, is associated with a significant decrease in sperm motility [27]. Negative effects on sperm functionality were also found following medium supplementation with other aromatic amino acids, as tryptophan and tyrosine, as well as with cysteine and methionine. These effects were, however, completely reversed following sperm exposure to either oviductal fluid or catalase [27]. It is important to highlight that, in mammals, the concentration of several amino acids differ in female reproductive fluids, as oviduct and uterus fluids, and blood serum [28]. Furthermore, dead sperm release the enzyme “amino acid oxidase,” which dehydrogenates and deaminates aromatic amino acids; this, in turn, causes the production of hydrogen peroxide and ammonia affecting sperm motility and survival [29].

The remaining amino acids identified in the donkey SP were alanine, glutamate, leucine, and the amino acid derivative, creatine. All of them showed a lower concentration in the azoo- than in normo-spermic individuals. They were correlated with sperm concentration and production as well as with each other and many other SP metabolites. However, they were not correlated with sperm kinetics. Alanine is a non-essential amino acid that at high concentrations affects sperm motility and decreases post-thaw sperm motility [30]. Glutamate is an excitatory amino acid sharing receptors with D-aspartate (for review see [31]). Recent studies have demonstrated that these receptors are expressed in rat and mouse testes and located in spermatogonia and Leydig cells [32]. Leucine is involved in cell calcium uptake that affects sperm motility, capacitation, and acrosome reaction [33]. Creatine is involved in cell energy metabolism. Two isoforms of creatine kinase have been found in chicken and human sperm as well as in human SP; experiments with metabolic blockers indicated a dependence of sperm motility on creatine kinase and phosphoryl creatine [34]. In mouse sperm, creatine sustained high ATP levels in sperm enhancing sperm capacitation and promoting sperm hyperactivation [35,36].

The other metabolites found in donkey SP, i.e., carnitine, choline, citrate, GPC, Myo-inositol, and TMAO may be included in a group of metabolic compounds. All these metabolites showed significantly lower values in the SP of the azoospermic stallion compared to those recorded in normospermic stallions. However, they are not able to discriminate higher and lower sperm motility individuals. Furthermore, the SP levels of these metabolites were not related to sperm kinematic parameters but correlated with sperm concentration and production. Their SP levels were correlated with each other and numerous other SP metabolites. In particular, carnitine is accumulated into the epididymis and sperm cells and involved in several metabolic pathways, as oxidative phosphorylation as well as in the transport of long-chain fatty acid into mitochondria for energy production, in removing metabolic by-products from cells, and in buffering the intra-mitochondrial ratio of free and acetylated co-enzyme A [37]. Besides, as ALC, it represents a ready energy source for spermatozoa and is involved in the transport of fatty acids into the mitochondria [38]. In the horse, a positive correlation has been found between SP L-carnitine levels with both sperm concentration and acrosomal membrane integrity. Furthermore, L-carnitine addition to semen extender preserved the motility of equine sperm stored at 5 °C [39]. Choline is an essential nutrient that may be converted into glycine and GPC [40] and is involved in the synthesis of the cell membrane, acetylcholine, and S-adenosylmethionine. The latter is a universal methyl donor and precursor of homocysteine, participating in the 1-carbon cycle, a pathway involved in sperm functionality [41]. Citrate is an intermediate product of the Krebs cycle and involved in ATP production [42]. In SP, it chelates calcium ions and limits sperm capacitation and spontaneous acrosome reaction [43]. High citrate concentration in SP was measured in infertile men although did not correlate with fertility [44]. GPC is produced by epididymal cells and released in the epididymal lumen; spermatozoa increase their GPC levels at maturation (for review see [45]). Myo-inositol is a carbocyclic sugar involved in cell signal transduction, antioxidant defenses, and osmoregulation; as phosphatidylinositol or as mono or polyphosphates works as a second messenger in many intracellular signal transduction pathways [46]. It is associated with an improved mitochondrial morphology [47] and membrane potential [48]. In bovine, the addition of 10 mM myo-inositol to sperm extender significantly increased total and progressive sperm motility in frozen-thawed semen after 4 h incubation [19]. In humans, using the above myo-inositol concentration, an increase of sperm motility was described after 2 h incubation in oligoasthenoteratospermic but not in normospermic patients [48]. Moreover, in human and bovine spermatozoa, the myo-inositol treatment improved MMP on the first hour of incubation [19,48]. TMAO is involved in several metabolic processes as well as in several metabolic diseases [49]. In human follicular fluid, TMAO levels are negatively related to embryo quality [50]. In a study on the effects of reactive oxygen species (ROS) levels on SP, metabolite milieu, and sperm dysfunction in normospermic men, TMAO showed a high upregulated metabolic fingerprinting and a positive correlation with ROS levels [51].

In the present study, carrying out analyses on semen samples obtained by repeated collections in the same animals, we found a lower variability within individuals than between individuals in terms of SP metabolite concentrations. This makes us believe that the values obtained in the individual samples are not random but typical of individuals. Using this methodological approach, individual reproductive characteristics can be discriminated. However, the lack of relevant differences of fertility among the evaluated stallions does not allow to obtain definitive results.

Interfacing all the information related to any single SP metabolite to the seminal parameters and the sperm kinetics by multivariate analysis, we obtained a single dot representing the single semen sample. Using a combined approach based on NMR data and PLS-DA features, we were able to obtain separate clusters for the different animals according to sperm motility (i.e., high and low sperm motility), with an independent cluster represented by the azoospermic stallion. Similar clusters have been developed in humans and animals to discriminate oligospermic from normospermic individuals [15,52] or high- and low-fertility males [16] and aimed to develop sophisticated algorithms capable of estimating male fertility with ever greater precision. The motility-associated metabolites with VIP scores > 1.0 slightly differed between the two models analyzed including either only the SP metabolites or the SP metabolites, semen and sperm kinematic parameters. Carnitine, phenylalanine, citrate, choline, and leucine showed similar features between the two models whereas acetate, lactate, and TMAO showed VIP scores > 1 in the first model and VIP < 1 in the second model. These results are in agreement with the findings in cattle [16] in which both citrate and leucine have been recognized as fertility-associated markers with high VIP scores. In a recent study in humans [53], citrate and choline SP values, together with lactate, lysine, arginine, valine, glutamine, creatinine, α-ketoglutaric acid, spermine, putrescine, and tyrosine, were found to differ significantly between patients with idiopathic oligoasthenoteratozoospermia and normospermic individuals and showed VIP scores > 1.

## 5. Conclusions

^1^H NMR spectroscopy allowed to identify 17 metabolites in the SP of donkey stallions. The concentrations of many of these metabolites varied significantly between azoospermic and normospermic stallions. Among them, only phenylalanine was able to differentiate stallions with higher and lower sperm motility and to achieve significant correlation coefficients with all sperm kinematic parameters. Some of these metabolites correlated with each other, and with semen quality parameters, such as sperm concentration. Other metabolites resulted to be independent. The multivariate analysis allowed to group all the individual information into a single dot whose spatial distribution assembled distinct clusters for stallions with different sperm motility, as well as for the azoospermic individual. This information provided interesting insights for the study of sperm metabolism and the development of an optimized semen extender in this species.

## Figures and Tables

**Figure 1 animals-11-00201-f001:**
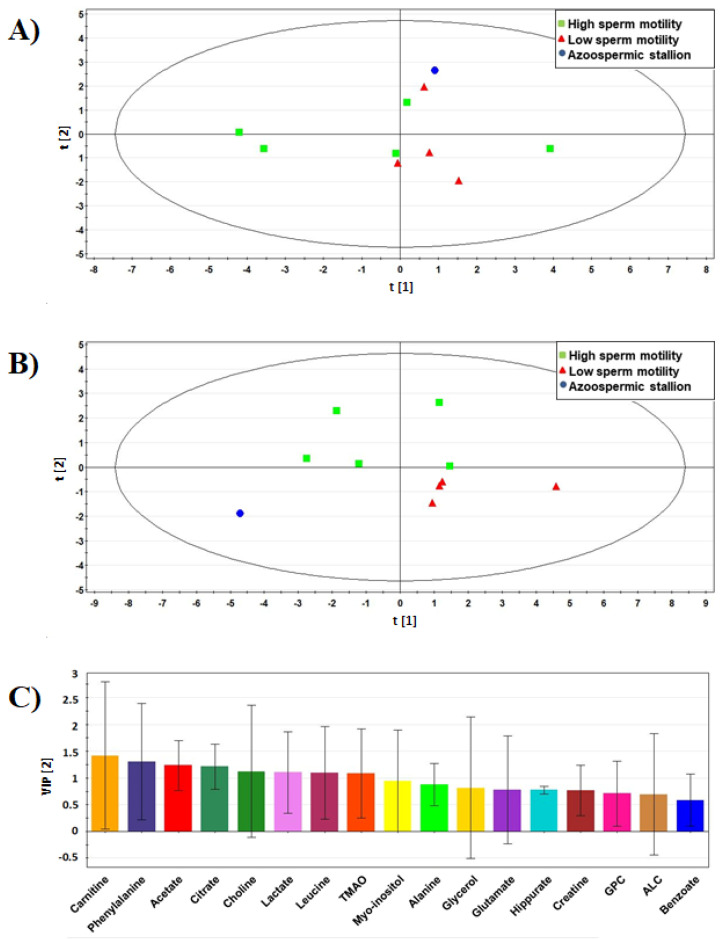
Multivariate analyses of seminal plasma metabolites of the 10 donkey stallions selected for the study and discriminated for sperm motility rate. (**A**) PCA score scatter plot from the ^1^H-NMR spectral data. PCA resulted in a two-component model with an R^2^X of 0.822 and a Q^2^ of 0.369. (**B**) PLS-DA score plot obtained from the same NMR data. The values of this two-component model were R^2^X = 0.567; R^2^Y = 0.652; Q^2^ = 0.18. In both score plots, green boxes represent stallions with high sperm motility (*n* = 5); red triangles represent stallions with lower sperm motility (*n* = 4); a dot represent the azoospermic stallion (*n* = 1). (**C**) Variable importance in projection (VIP) of the NMR variables for the model reported in B.

**Figure 2 animals-11-00201-f002:**
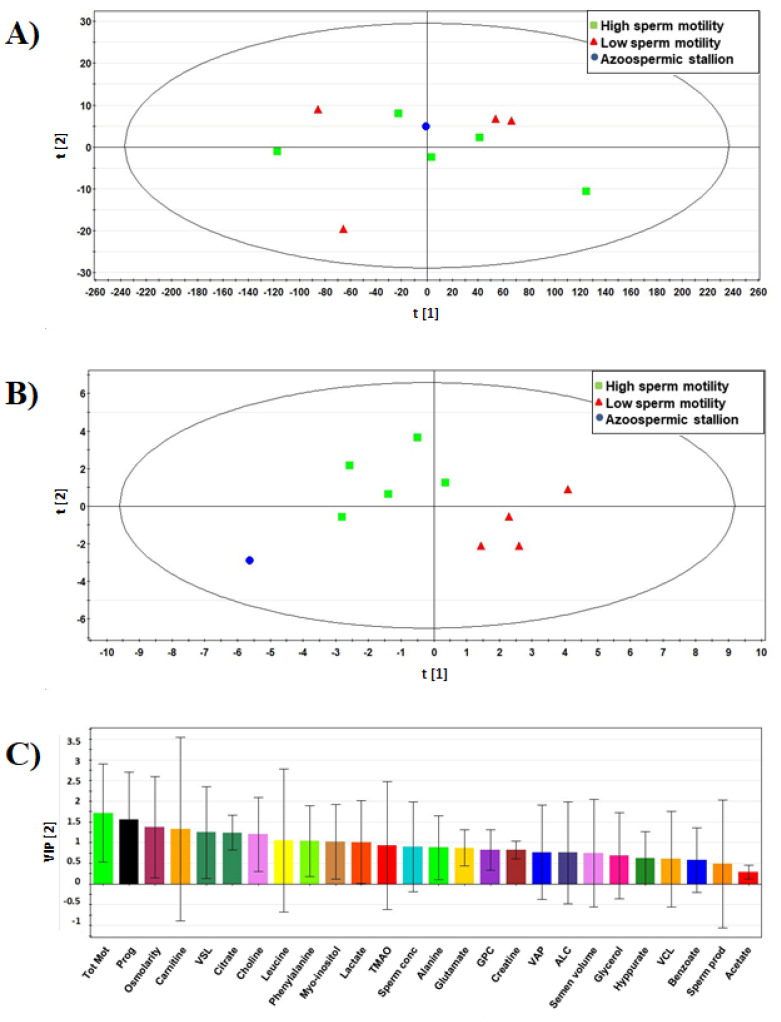
Multivariate analyses of seminal plasma metabolites, semen parameters and sperm kinetics in donkey stallions (*n* = 10). (**A**) PCA score scatter plot from both the ^1^H-NMR spectral data and data of sperm quality. PCA resulted in a two-component model with an R^2^X value of 0.992 and a Q^2^ of 0.287. (**B**) PLS-DA score plot obtained from the same dataset. The values of this two-component model were R^2^X = 0.49; R^2^Y = 0.737; Q^2^ = 0.265. In both score plots, green boxes represent stallions with high sperm motility (*n* = 5); red triangles represent stallions with lower sperm motility (*n* = 4); a dot represents the azoospermic stallion (*n* = 1). (**C**) Variable importance in projection (VIP) of the variables for model reported in B. Tot Mot (total sperm motility), Prog (progressive sperm motility), VCL (curvilinear velocity), VSL (straight-line velocity), VAP (average path velocity), Sperm prod (sperm production), GPC (glycerophosphocholine), ALC (O-acetyl carnitine), TMAO (trimethylamine N-oxide).

**Table 1 animals-11-00201-t001:** Mean (±SD) values of seminal plasma metabolites (mM), semen parameters, and sperm kinetics in Martina Franca donkey stallions (*n* = 7).

Seminal Plasma Metabolites	Sperm Motility		Azoospermic
	High	Low	
Acetate	0.153 ± 0.047	0.192 ± 0.030	0.127 ± 0.045
Alanine	0.043 ± 0.259 ^a^	0.039 ± 0.268 ^a^	0.003 ± 0.001 ^b^
Benzoate	0.013 ± 0.013	0.017 ± 0.006	0.007 ± 0.010
Carnitine	0.420 ± 0.115 ^A^	0.390 ± 0.074 ^A^	0.012 ± 0.017 ^B^
Choline	0.260 ± 0.024 ^a^	0.413 ± 0.139 ^A^	0.037 ± 0.012 ^Bb^
Citrate	0.069 ± 0.029 ^AB^	0.099 ± 0.021 ^A^	0.023 ± 0.005 ^B^
Creatine	0.356 ± 0.160 ^a^	0.400 ± 0.171 ^a^	0.006 ± 0.002 ^b^
Glutamate	1.815 ± 0.811 ^a^	2.362 ± 0.558 ^A^	0.141 ± 0.027 ^Bb^
Glycerol	18.023 ± 8.096	19.066 ± 1.650	21.928 ± 0.197
GPC	1.491 ± 0.852 ^a^	1.935 ± 0.180 ^A^	0.074 ± 0.004 ^Bb^
Hippurate	0.002 ± 0.003	0.008 ± 0.007	0.004 ± 0.006
Lactate	0.589 ± 0.189 ^a^	0.781 ± 0.053 ^A^	0.250 ± 0.028 ^Bb^
Leucine	0.043 ± 0.024 ^a^	0.039 ± 0.007 ^a^	0.003 ± 0.001 ^b^
Myo-inositol	0.340 ± 0.259 ^ab^	0.603 ± 0.178 ^a^	0.040 ± 0.008 ^b^
ALC	0.107 ± 0.028	0.199 ± 0.145	0.019 ± 0.001
Phenylalanine	0.009 ± 0.005 ^a^	0.001 ± 0.002 ^b^	0.000 ± 0.000 ^b^
TMAO	0.348 ± 0.270	0.229 ± 0.050	0.001 ± 0.000
Semen parameters			
pH	7.50 ± 0.08 ^A^	7.66 ± 0.19 ^A^	8.45 ± 0.06 ^B^
Osmolarity (mOsm)	282 ± 8.185	293 ± 11	293 ± 8
Volume (mL)	48.3 ± 7.638 ^a^	46.7 ± 18.9 ^a^	17.5 ± 3.5 ^b^
Sperm concentration (10^6^/mL)	320.7 ± 19.3	333.9 ± 3.7	0
Sperm production (10^9^)	15.3 ± 2.5	15.7 ± 6.7	0
Sperm kinetics at time 0			
Tot Mot (%)	93.1 ± 6.1 ^a^	56.3 ± 17.0 ^b^	0
Prog (%)	33.9 ± 6.0 ^A^	17.7 ± 3.2 ^B^	0
VCL (µm/s)	227.3 ± 25.2	205.9 ± 31.5	0
VSL (µm/s)	87.0 ± 13.4	70.8 ± 7.4	0
VAP (µm/s)	123.5 ± 25.5	100.6 ± 7.4	0
Sperm kinetics after 24 h storage			
Tot Mot (%)	64.3 ± 23.4 ^A^	2.7 ± 3.8 ^B^	0
Prog (%)	21.3 ± 14.0 ^a^	0.5 ± 0.0 ^b^	0
VCL (µm/s)	184.1 ± 27.8 ^A^	73.8 ± 7.4 ^B^	0
VSL (µm/s)	74.6 ± 20.3 ^A^	21.1 ± 8.1 ^B^	0
VAP (µm/s)	97.1 ± 22.2 ^A^	33.3 ± 9.3 ^B^	0

Data are referred to single semen collections in six normospermic stallions and a double semen sample collected on a monthly basis in the azoospermic stallion. These stallions were discriminated on the basis of total sperm motility as high motility (*n* = 3), low motility (*n* = 3) and azoospermic (*n* = 1) individuals. SP metabolites have been evaluated by ^1^H NMR spectroscopy. Sperm kinetics were evaluated at the time of semen collection (T0) and after 24 h storage at 4 °C (T24). GPC (Glycerophosphocholine), ALC (O-Acetylcarnitine), TMAO (Trimethylamine N-oxide), Tot Mot (total sperm motility), Prog (progressive sperm motility), VCL (curvilinear velocity), VSL (straight-line velocity) and VAP (average path velocity). Different letters in the same row indicate significant differences (A, B; *p* < 0.01) (a, b; *p* < 0.05).

**Table 2 animals-11-00201-t002:** Mean (±SD) values of seminal plasma metabolites (mM), semen parameters, sperm kinetics, mitochondrial membrane potential (MMP) and oxidative status recorded in the three donkey stallions (MF1, MF2 and RO) submitted to three sperm collections repeated on a monthly basis.

Seminal Plasma Metabolites	Stallions		
	MF1	MF2	RO
Acetate	0.134 ± 0.060	0.084 ± 0.011	0.152 ± 0.055
Alanine	0.200 ± 0.074	0.168 ± 0.018	0.150 ± 0.063
Benzoate	0.032 ± 0.017	0.078 ± 0.076	0.035 ± 0.013
Carnitine	0.284 ± 0.105	0.237 ± 0.028	0.191 ± 0.072
Choline	0.251 ± 0.105	0.178 ± 0.028	0.165 ± 0.017
Citrate	0.092 ± 0.092	0.043 ± 0.021	0.094 ± 0.042
Creatine	0.088 ± 0.013	0.098 ± 0.020	0.091 ± 0.032
Glutamate	0.917 ± 0.364 ^a^	0.539 ± 0.100 ^b^	0.447 ± 0.161 ^b^
Glycerol	17.851 ± 1.754 ^a^	10.893 ± 2.146 ^b^	15.265 ± 5.009 ^ab^
GPC	0.546 ± 0.164	0.581 ± 0.164	0.459 ± 0.209
Hippurate	0.005 ± 0.009 ^A^	0.048 ± 0.015 ^B^	0.011 ± 0.014 ^A^
Lactate	0.349 ± 0.142 ^A^	0.297 ± 0.059 ^A^	2.226 ± 1.193 ^B^
Leucine	0.042 ± 0.020 ^a^	0.020 ± 0.005 ^b^	0.026 ± 0.011 ^ab^
Myo-Inositol	0.262 ± 0.127	0.308 ± 0.038	0.213 ± 0.106
ALC	0.049 ± 0.012	0.045 ± 0.011	0.032 ± 0.009
Phenylalanine	0.005 ± 0.005	0.018 ± 0.016	0.008 ± 0.006
TMAO	0.119 ± 0.018	0.110 ± 0.025	0.145 ± 0.042
Semen parameters			
pH	7.64 ± 0.56 ^a^	7.52 ± 0.14 ^a^	8.18 ± 0.23 ^b^
Osmolarity (mOsm)	296 ± 10 ^a^	293 ± 1 ^A^	314 ± 11 ^Bb^
Semen volume (mL)	23.7 ± 4.7 ^a^	69.0 ± 33.0 ^b^	65.2 ± 16.4 ^b^
Sperm concentration (10^6^/mL)	229.0 ± 54.1 ^a^	182.0 ± 43.0 ^ab^	129.5 ± 29.6 ^b^
Sperm production (10^9^)	5.25 ± 0.31	13.32 ± 8.8	8.15 ± 1.23
Sperm kinetics at time 0			
Tot Mot (%)	80.5 ± 14.8	90.1 ± 8.2	69.1 ± 13.6
Prog (%)	33.2 ± 8.0	25.1 ± 8.5	22.8 ± 1.4
VCL (µm/s)	110.0 ± 7.1 ^a^	127.7 ± 7.9 ^b^	117.7 ± 0.1 ^ab^
VSL (µm/s)	58.3 ± 6.5	59.8 ± 9.5	48.5 ± 2.4
VAP (µm/s)	77.8 ± 10.5 ^a^	95.9 ± 6.7 ^b^	82.0 ± 10.1 ^ab^
MMP and oxidative status at time 0			
MMP (FoB/FoA)	49.8 ± 17.3	79.6 ± 24.7	78.8 ± 20.9
LPO (FoA/(FoA+FoB))	19.3 ± 8.4	14.8 ± 1.8	14.9 ± 4.2
Anti-LPO (FoB/(FoA+FoB))	30.1 ± 9.6	27.9 ± 7.3	24.3 ± 10.1
NBT (Formazan µg)	7.78 ± 2.77	11.52 ± 5.44	5.80 ± 2.80
Sperm kinetics after 24 h storage			
Tot Mot (%)	75.5 ± 5.3 ^a^	63.5 ± 11.5 ^ab^	40.5 ± 20.9 ^b^
Prog (%)	33.5 ± 2.5 ^A^	22.8 ± 5.9 ^AB^	10.4 ± 10.7 ^B^
VCL (µm/s)	110.5 ± 6.5	101.8 ± 9.4	98.7 ± 23.9
VSL (µm/s)	65.5 ± 4.7	50.9 ± 4.4	47.3 ± 24.9
VAP (µm/s)	85.0 ± 6.5	71.8 ± 9.8	71.1 ± 32.8
MMP and oxidative status after 24 h storage			
MMP (FoB/FoA)	40.8 ± 25.3	75.1 ± 25.9	49.4 ± 15.5
LPO (FoA/(FoA+FoB))	24.2 ± 15.3	14.7 ± 4.1	13.9 ± 5.1
Anti-LPO (FoB/(FoA+FoB))	35.1 ± 12.0	27.5 ± 4.3	32.6 ± 8.5
NBT (Formazan µg)	7.73 ± 3.18	7.88 ± 5.11	7.44 ± 3.12

MF (Martina Franca breed) and RO (Romanian breed) stallions. Seminal plasma metabolites were evaluated by ^1^H NMR spectroscopy and their concentrations are expressed in mM. Sperm kinetics were evaluated at the time of semen collection and after 24 h storage at 4 °C. GPC (Glycerophosphocholine), ALC (O-Acetylcarnitine), TMAO (Trimethylamine N-oxide), Tot Mot (total sperm motility), Prog (progressive sperm motility), VCL (curvilinear velocity), VSL (straight-line velocity) and VAP (average path velocity). LPO (lipid peroxidation), anti-LPO (anti-peroxidation activity) and NBT (nitroblue tetrazolium assay). Different letters in the same row indicate significant differences (A, B; *p* < 0.01) (a, b; *p* < 0.05).

**Table 3 animals-11-00201-t003:** Correlation coefficients (R) between seminal plasma metabolites evaluated by ^1^H NMR spectroscopy in ten donkey stallions.

	Alanine	Carnitine	Choline	Citrate	Creatine	Glutamate	GPC	Leucine	Myo-Inositol	ALC	TMAO
Acetate	+0.514	+0.396	+0.557	+0.745 **	+0.469	+0.596 *	+0.571	+0.414	+0.335	+0.559	+0.083
Alanine		+0.923 **	+0.784 **	+0.544	+0.969 **	+0.938 **	+0.751 **	+0.852 **	+0.696 *	+0.754 **	+0.709 **
Carnitine			+0.776 **	+0.622 *	+0.873 **	+0.874 **	+0.763 **	+0.907 **	+0.658 *	+0.606*	+0.719 **
Choline				+0.697 *	+0.781 **	+0.767 **	+0.650 *	+0.652 *	+0.902 **	+0.852**	+0.465
Citrate					+0.541	+0.600 **	+0.505	+0.707 **	+0.513	+0.450	+0.345
Creatine						+0.908 **	+0.666 *	+0.847 **	+0.781 **	+0.759 **	+0.776 **
Glutamate							+0.890 **	+0.769 **	+0.625 *	+0.755 **	+0.527
GPC								+0.562	+0.421	+0.580	+0.276
Leucine									+0.614 *	+0.416	+0.845 **
Myo-inositol										+0.746 **	+0.623 *
ALC											+0.262

This table refers only to those metabolites showing more than one statistically significant correlation with the others. GPC (glycerophosphocholine), ALC (O-acetyl carnitine), TMAO (trimethylamine N-oxide). * (*p* < 0.05), ** (*p* < 0.01).

**Table 4 animals-11-00201-t004:** Correlation coefficients (R) between seminal plasma metabolites and semen parameters in donkey stallions (*n* = 10).

	Semen Parameters		
SP Metabolites	Semen Volume	Sperm Concentration	Sperm Production
Acetate	+0.182	+0.180	+0.476
Alanine	+0.249	+0.759 **	+0.867 **
Benzoate	+0.752 **	+0.048	+0.050
Carnitine	+0.432	+0.886 **	+0.850 **
Choline	+0.345	+0.732 **	+0.700 **
Citrate	+0.575	+0.424	+0.593 *
Creatine	+0.314	+0.751 **	+0.918 **
Glutamate	+0.223	+0.770 **	+0.869 **
Glycerol	−0.484	−0.314	−0.105
GPC	+0.144	+0.746 **	+0.712 **
Hippurate	+0.581	−0.105	+0.074
Lactate	+0.320	−0.062	−0.068
Leucine	+0.485	+0.703 **	+0.812 **
Myo-inositol	+0346	+0.692 *	+0.687 *
ALC	+0.155	+0.560	+0.632 *
Phenylalanine	+0.540	+0.224	−0.109
TMAO	+0.335	+0.628	+0.750 **
Semen parameters			
Osmolarity	−0.066	−0.482	−0.324
pH	−0.078	−0.384	−0.073
Sperm volume		−0.396	+0.797 **
Sperm concentration			+0.228

SP metabolites have been evaluated by ^1^H NMR spectroscopy. GPC (glycerophosphocholine), ALC (O-acetyl carnitine), TMAO (trimethylamine N-oxide). * (*p* < 0.05), ** (*p* < 0.01).

**Table 5 animals-11-00201-t005:** Correlation coefficients (R) between seminal plasma metabolites and sperm kinematic parameters evaluated in the nine donkey stallions at the time of semen collection and after 24 h of storage at 4 °C.

		Sperm Kinematic Parameters at Time 0				Sperm Kinematic Parameters after 24 h of Storage				
Semen Plasma Metabolites	*Tot* *Mot*	*Prog*	*VCL*	*VSL*	*VAP*	*Tot* *Mot*	*Prog*	*VCL*	*VSL*	*VAP*
Acetate	−0.340	−0.744 *	−0.045	−0.593	−0.067	−0.434	−0.508	−0.144	−0.161	−0.098
Benzoate	−0.029	−0.095	+0.556	+0.441	+0.689 *	+0.392	+0.323	+0.645*	+0.473	+0.534
Citrate	−0.341	−0.760 **	+0.278	−0.378	+0.221	+0.023	−0.032	+0.386	+0.304	+0.380
Phenylalanine	+0.326	+0.517	+0.226	+0.565	+0.558	+0.821 **	+0.853 **	+0.847**	+0.902 **	+0.909 **
Semen parameters										
Osmolarity	−0.666 *	−0.532	−0.120	−0.025	+0.106	−0.249	−0.273	−0.058	−0.184	−0.141
Semen volume	+0.082	−0.136	+0.549	+0.470	+0.736 *	+0.425	+0.375	+0.591 *	+0.484	+0.555
Sperm kinematic parameters										
Tot Mot		+0.742 *	+0.599	+0.476	+0.457		+0.951 ^**^	+0.757 **	+0.900 **	+0.892 **
Prog			+0.183	+0.507	+0.156			+0.867 **	+0.916 **	+0.886 **
VCL				+0.332	+0.668 *				+0.966 **	+0.975 **
VSL					+0.783 **					+0.994 **

SP metabolites were evaluated by ^1^H NMR spectroscopy. This table reports only parameters showing significant correlations. Tot Mot (total sperm motility), Prog (progressive sperm motility), VCL (curvilinear velocity), VSL (straight-line velocity) and VAP (average path velocity). * (*p* < 0.05), ** (*p* < 0.01).

## Data Availability

The data presented in this study are available on request from the corresponding author.

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
