# Peer review of "Relationships between Seminal Plasma Metabolites, Semen Characteristics and Sperm Kinetics in Donkey (Equus asinus)"

_animals, 2021, doi:10.3390/ani11010201_

Round 1

Reviewer 1 Report

The study analyzed ejaculates collected from male donkeys to identify potential metabolic parameters that would be indicative of semen quality to be used for AI purposes.  The study was comprehensive and provided data base information that can be utilized in the industry to assist in the assessment of semen quality.  It would be interesting to apply the information to many other species for which AI is the method of breeding.

Author Response

We are sincerely grateful to the reviewer for his/her appreciation for our study

Reviewer 2 Report

General comments: This is a very interesting study in which authors explored the seminal plasma metabolic components and its relationship with sperm characteristics in donkey. Moreover, the fact that this study is the first of its kind in this species, which is currently at risk of extinction , makes it even more important. Overall, the Introduction section is well defined and straightforward, with a well-defined goal. The same goes for the Materials and methods section, with a complete statistical analysis, which is welcomed, taking into consideration the low number of animals that could be enrolled in the study. However, the Results part of this manuscript requires extensive revision since not all the information given in the tables matches what is said in the text. The same applies to the Discussion section which, besides being far too long, does not offer a proper discussion of the results and is not relevant for the purpose of this research.

I have taken the liberty to do some editing of the English and text format on an attached draft. Some sentences are not clear because of the language used. Before the authors submit the revised version of their paper they would be wise to get a native English speaker to read through the article

Title: It is suggested that the title be written in such a way that is expresses greater clarity according to the objective. It does not give any proper information as it is right now.

Materials and methods:

Line 142: What media?

Line 148: In section 2.6 (MMP) nothing is said regarding the chemical inhibitor CCCP, which is listed in the reagents section. Did authors use is as a positive control? If not, please remove it.

Line 153: After staining, why did authors incubate samples for an additional 30 min? Moreover, and just out of curiosity, did authors have any issue dissolving JC-1 in the medium prior to staining?

Results:

Line 248: What about pH? It was significantly higher in the azoospermic individual.

Line 257: In the experimental design section it is said that AI was carried out to analyze fertility data from the stallions enrolled in this study, but here it is said that 17 and 29 animals were used. Please correct, since only the donkeys from which seminal plasma was collected are relevant in this study.

Line 261: Table 1, please include the units of measurement (mM) also in the table, not just in the information above. Also, ALC, GPC and TMAO should also be used since authors also use other abbreviations such as VCL, VAP, etc. Furthermore, superscript letters from citrate and myo-inositol are missing in the high sperm motility column.

Line 262: For the azoospermic individual, it is claim that semen collection was repeated after one month. Therefore, are the results showed in Table 1 the average from these 2 collections? It is not clear.

Line 271: Section 3.2. This section is very confusing. Up to this point it is claim that 3 stallions were submitted for repeated semen collections (n=3) “for evaluating the metabolite variability within the same individual”. However, the information given here is the average results obtained from these repeated collections? For clarification, are authors trying to compare intraindividual SP characteristics or interindividual differences from 2 different breeds? Either way, this whole part must be corrected.

Line 274: How is R the stallion with lower sperm motility? In the table there are no differences regarding this parameter between R and MF individuals. Also, it is claimed that “R showed a lower glutamate content and a lower sperm concentration…than the other 2 stallions”, but glutamate did not show differences between MF2 and R, and superscript letter is missing from the sperm concentration of MF2 column so one can only guess which one is right.

Line 277: Cannot state “sperm kinetic characteristics at T0 were accentuated at T24 in both total and progressive motility” because at T0 there are no differences between individuals.

Line 281: Results regarding MMP, LPO, anti-LPO and NBT of stored semen samples is already in Table 2. Why did authors say it is not in tables? And where and from what samples did those rates come from?

Line 285: Table 2. Superscript letters from Hippurate, please correct. Definition of what sperm production is should be stated in the methods section.

Line 295: Apart from the metabolites that had no association with the others (benzoate, glycerol, hippurate and phenylalanine), the ones with significant correlation coefficients should also be named here (since those are the ones that really stand out).

Line 300: Table 3. Please improve the format of this table so that words do not separate.

Line 305: Authors claim “correlations between metabolites and typical seminal parameters, such as osmolarity, volume, sperm concentration and production” but in Table 4 the semen parameters columns are volume, concentration and production. Also, is Sperm production missing from the first column?

Line 310: Please specify which associations those were.

Line 349: This sentence should be put before talking about C.

Discussion:

Overall, the information given regarding each metabolite is interesting, but not very useful for the purpose of this study. As a first exploration, I understand that full in-depth conclusions would be hard to present, but it would be expected that the discussion of results was done in a broader context by the large number of variables evaluated and the relevance of the results found. As said in the manuscript, the results obtained in this study could really “offer insights for future technological applications, for the study of sperm metabolism and the development of optimized semen extenders in this species”. For that reason, even if I agree with a brief explanation of each metabolite, the discussion part should focus on these important aspects. This is especially interesting regarding the information acquired from the correlation and multivariate analyses (what about the VIP scores?), which are barely discussed in this section.

Final remarks: At this point, even though I am generally positive about this manuscript, it cannot be published as it is now. This is certainly a very interesting work, which was executed well and offers interesting information. However, there is room for improvement and therefore correct revision is paramount.

Author Response

General comments: This is a very interesting study in which authors explored the seminal plasma metabolic components and its relationship with sperm characteristics in donkey. Moreover, the fact that this study is the first of its kind in this species, which is currently at risk of extinction, makes it even more important. Overall, the Introduction section is well defined and straightforward, with a well-defined goal. The same goes for the Materials and methods section, with a complete statistical analysis, which is welcomed, taking into consideration the low number of animals that could be enrolled in the study. However, the Results part of this manuscript requires extensive revision since not all the information given in the tables matches what is said in the text. The same applies to the Discussion section which, besides being far too long, does not offer a proper discussion of the results and is not relevant for the purpose of this research.

I have taken the liberty to do some editing of the English and text format on an attached draft. Some sentences are not clear because of the language used. Before the authors submit the revised version of their paper they would be wise to get a native English speaker to read through the article

R: We greatly appreciate your positive and constructive comments as well as your help in editing and improving our manuscript. We have done our best to modify the paper based on your guidance.

 Title: It is suggested that the title be written in such a way that is expresses greater clarity according to the objective. It does not give any proper information as it is right now.

R: We changed the title.

Materials and methods:

Line 142: What media?

R: We added the medium used.

Line 148: In section 2.6 (MMP) nothing is said regarding the chemical inhibitor CCCP, which is listed in the reagents section. Did authors use is as a positive control? If not, please remove it.

R: We removed CCCP.

Line 153: After staining, why did authors incubate samples for an additional 30 min? Moreover, and just out of curiosity, did authors have any issue dissolving JC-1 in the medium prior to staining?

R: The evaluation of the MMP and LPO is included within a panel of evaluations of sperm function that we developed in different animal species. Some of these assessments use fluorochromes that enter into the spermatozoon via an Acetyl-Methyl (AM) ester group and an additional 30 min of incubation is required to allow de-esterification of this carrier. In this study, we have respected the times that we used for the developed protocol. JC-1 was diluted in DMSO and stored at -80 °C into dark boxes. In the past (Boni et al., MRD, 2016; Boni et al., Andrology, 2017), we have sometimes found JC-1 precipitates in the sample. In that case, we solved the problem with a discontinuous Percoll gradient. This event did not occur in our recent analyses.

Results:

Line 248: What about pH? It was significantly higher in the azoospermic individual.

R: We added this information in the Results section.

Line 257: In the experimental design section it is said that AI was carried out to analyze fertility data from the stallions enrolled in this study, but here it is said that 17 and 29 animals were used. Please correct, since only the donkeys from which seminal plasma was collected are relevant in this study.

R: We are sorry for this confusion; 17 and 29 were referred to the inseminated mares (total=46). We added this information in the text.

Line 261: Table 1, please include the units of measurement (mM) also in the table, not just in the information above. Also, ALC, GPC and TMAO should also be used since authors also use other abbreviations such as VCL, VAP, etc. Furthermore, superscript letters from citrate and myo-inositol are missing in the high sperm motility column.

R: We used abbreviations also for ALC, GPC, and TMAO and added this information in the Table 1 legend; In addition, we added the mM unit in Table 1and superscript letters from citrate and myo-inositol.

Line 262: For the azoospermic individual, it is claim that semen collection was repeated after one month. Therefore, are the results showed in Table 1 the average from these 2 collections? It is not clear.

R: We are sorry for this confusion; we added this information in Table 1 legend.

Line 271: Section 3.2. This section is very confusing. Up to this point it is claim that 3 stallions were submitted for repeated semen collections (n=3) “for evaluating the metabolite variability within the same individual”. However, the information given here is the average results obtained from these repeated collections? For clarification, are authors trying to compare intraindividual SP characteristics or interindividual differences from 2 different breeds? Either way, this whole part must be corrected.

R: We clarified the point adding more information in Table 2 legend.

Line 274: How is R the stallion with lower sperm motility? In the table there are no differences regarding this parameter between R and MF individuals. Also, it is claimed that “R showed a lower glutamate content and a lower sperm concentration…than the other 2 stallions”, but glutamate did not show differences between MF2 and R, and superscript letter is missing from the sperm concentration of MF2 column so one can only guess which one is right.

R: Sorry, we referred to Experiment 1 in which we used 70% TotMot as the discriminant threshold between stallions with higher and lower motility rate. We removed this reference and modified the text accordingly.

Line 277: Cannot state “sperm kinetic characteristics at T0 were accentuated at T24 in both total and progressive motility” because at T0 there are no differences between individuals.

R: You are right: it was just a tendency. We modified this sentence accordingly.

Line 281: Results regarding MMP, LPO, anti-LPO and NBT of stored semen samples is already in Table 2. Why did authors say it is not in tables? And where and from what samples did those rates come from?

R: These results were derived from a new elaboration of the data to evaluate, independently by individuals, if these parameters were time-dependent. We modified the text accordingly.

Line 285: Table 2. Superscript letters from Hippurate, please correct. Definition of what sperm production is should be stated in the methods section.

R: Done

Line 295: Apart from the metabolites that had no association with the others (benzoate, glycerol, hippurate and phenylalanine), the ones with significant correlation coefficients should also be named here (since those are the ones that really stand out).

R: We modified the text accordingly.

Line 300: Table 3. Please improve the format of this table so that words do not separate.

R: Done. Unfortunately, the Table contains a lot of (17 x 17) parameters (in truth 16 because one would have to confront himself). This brings the Table out of the narrow template. It would be necessary to turn the Table 90 °. I cannot do it with Word; however, I might transform the Table into a Figure and turn it 90 °. I hope that the Editorial staff can do this or accept a small leap of the Table from the imposed limits.

Line 305: Authors claim “correlations between metabolites and typical seminal parameters, such as osmolarity, volume, sperm concentration and production” but in Table 4 the semen parameters columns are volume, concentration and production. Also, is Sperm production missing from the first column?

R: We moved the Table reference at the end of this paragraph.

Line 310: Please specify which associations those were.

R: We replaced “associations” with “correlations”.

Line 349: This sentence should be put before talking about C.

R: We moved the sentence before C, as requested.

Discussion:

Overall, the information given regarding each metabolite is interesting, but not very useful for the purpose of this study. As a first exploration, I understand that full in-depth conclusions would be hard to present, but it would be expected that the discussion of results was done in a broader context by the large number of variables evaluated and the relevance of the results found. As said in the manuscript, the results obtained in this study could really “offer insights for future technological applications, for the study of sperm metabolism and the development of optimized semen extenders in this species”. For that reason, even if I agree with a brief explanation of each metabolite, the discussion part should focus on these important aspects. This is especially interesting regarding the information acquired from the correlation and multivariate analyses (what about the VIP scores?), which are barely discussed in this section.

R: We operated many modifications in the Discussion, reducing the amount of information on individual metabolites, and focusing on the objectives of the study. However, this is the first study on this matter in this species which does not have large numbers and does not make extensive use of instrumental insemination. This made it very difficult to radicalize the research, as in the bovine species (Kumar et al., 2015), where two well-differentiated fertility groups were compared. In our case, we started from laboratory data on sperm motility rate and, fortunately, thanks to a project for the production of mules, we have obtained the fertility of these animals divided into groups arranged on the motility rate. Unfortunately, the fertility data did not support much of our grouping rationale. However, the animals classified with lower motility while showing at T0 fertility similar to that of the high motility group then proved to be bad coolers with a collapse of sperm motility at T24. Finally, we believe that, despite limitations linked to the availability of the material, this study provides a lot of unpublished and interesting scientific material also for comparative purposes.

Reviewer 3 Report

I think the sentence in lines: 63=67 is too complicated for the reader.

line 261: For a clear description of Table 1 it would be nice to add the total number of stallions at this stage of the experiments (n =?) and in the line: 284 i 285, the same.

Line 313: donkey stallions (=10). Is it possible to put the letter "n" in the parenthesis?

Line 436-439: The second sentence does not confirm the first for me. If carnitine is untaken by sperm then how can it be involved in several metabolic pathways?

Author Response

We are sincerely grateful to the reviewer for his/her appreciation for our study.

I think the sentence in lines: 63=67 is too complicated for the reader.

R: We modified this sentence.

line 261: For a clear description of Table 1 it would be nice to add the total number of stallions at this stage of the experiments (n =?) and in the line: 284 i 285, the same.

R: Done

Line 313: donkey stallions (=10). Is it possible to put the letter "n" in the parenthesis?

R: Done

Line 436-439: The second sentence does not confirm the first for me. If carnitine is untaken by sperm then how can it be involved in several metabolic pathways?

R: We modified the sentence and added a new reference, i.e., a review article on this topic.

Round 2

Reviewer 2 Report

Overall, authors have greatly improve their manuscript (especially the discussion part), according to my previous comments and I am very positive about it. I also do agree with authors that it provides very interesting information in a species in which it is quite difficult to work with. However, in order to avoid confusion, further clarification of section 3.2 is needed. In general, my questions arise from the fact that the aim of this experiment was to "evaluate the metabolite variability within the same individual", but the information presented in table 2 is the average results (3 collections) from each of the 3 stallions (MF1, MF2 and R). I have commented on it more extensively down below.

Line 259: Table 1. Perhaps I did not explain myself properly. When I said that the units of measurement (in this case mM) should also be written in the table and not just the information above, with above I meant in the text just before the table. An extra column with the units is not needed, it could be perfectly written in the column title: "Seminal plasma metabolites (mM)".

Line 270: Section 3.2. This part has not been properly clarified. There is still no answer regarding my previous question: Are authors trying to compare intraindividual SP characteristics or interindividual differences from 2 different breeds? It is clear that 3 stallions (2 MF and 1 R) were submitted to 3 sperm collections, and the results showed in table 2 are the average of those 3 collections, but if the objective is to evaluate "the metabolite variability within the same individual", how can this be achieved by comparing the average results from each individual?

Line 273: Please correct because this sentence still does not match what is showed in table 2. Glutamate content and sperm concentration between MF2 and R are not different. Also, include the pH since it is significantly higher in R compared to MF1 and MF2.

Line 280: I am sorry but I still do not understand. What indidividual are those percentages referring to? The confusion is probably arising from the fact that the aim of this section (3.2) is not clear yet.

Line 284: Table 2. Same as in table 1. Maybe it was a misunderstanding, but the units of measurement were fine as they were written before (right after the parameter). Simply add the "mM" in the column title.

Line 285: Table 2. Superscript letters from Hippurate are still not correct. What is that "D" doing there?).

Line 312: Table 4. Please fix the format of this table.

Author Response

Overall, authors have greatly improve their manuscript (especially the discussion part), according to my previous comments and I am very positive about it. I also do agree with authors that it provides very interesting information in a species in which it is quite difficult to work with. However, in order to avoid confusion, further clarification of section 3.2 is needed. In general, my questions arise from the fact that the aim of this experiment was to "evaluate the metabolite variability within the same individual", but the information presented in table 2 is the average results (3 collections) from each of the 3 stallions (MF1, MF2 and R). I have commented on it more extensively down below.

R. We are sincerely grateful to the reviewer for the splendid work he/she has done which has greatly improved the quality of our manuscript. We make amends for the numerous errors and typos he/she had to correct in our manuscript.

Line 259: Table 1. Perhaps I did not explain myself properly. When I said that the units of measurement (in this case mM) should also be written in the table and not just the information above, with above I meant in the text just before the table. An extra column with the units is not needed, it could be perfectly written in the column title: "Seminal plasma metabolites (mM)".

R. Done

Line 270: Section 3.2. This part has not been properly clarified. There is still no answer regarding my previous question: Are authors trying to compare intraindividual SP characteristics or interindividual differences from 2 different breeds? It is clear that 3 stallions (2 MF and 1 R) were submitted to 3 sperm collections, and the results showed in table 2 are the average of those 3 collections, but if the objective is to evaluate "the metabolite variability within the same individual", how can this be achieved by comparing the average results from each individual?

R. We apologize for this unclear information. We did not intend to evaluate in this study the differences between Martina Franca vs. Romanian breeds. The repeated sampling was devised because these three stallions were housed close to our laboratory. Hence, we could repeat the sampling and add evaluations related to the sperm mitochondrial activity and oxidative status. Accidentally two of these stallions were MF and 1 RO. To respond to your right criticism, we have adopted a CV-ANOVA method, transforming the data into coefficients of variation, not considering the values ​​themselves but only the variation between and within individuals. We also included ANOVA for repeated measures as this is a more pertinent analysis for our experimental model. We checked the results obtained and, fortunately, we did not find any difference with the results already proposed.

Line 273: Please correct because this sentence still does not match what is showed in table 2. Glutamate content and sperm concentration between MF2 and R are not different. Also, include the pH since it is significantly higher in R compared to MF1 and MF2.

R.Done

Line 280: I am sorry but I still do not understand. What indidividual are those percentages referring to? The confusion is probably arising from the fact that the aim of this section (3.2) is not clear yet.

R. We removed the sentence.

Line 284: Table 2. Same as in table 1. Maybe it was a misunderstanding, but the units of measurement were fine as they were written before (right after the parameter). Simply add the "mM" in the column title.

R. Done

Line 285: Table 2. Superscript letters from Hippurate are still not correct. What is that "D" doing there?).

R. Done

Line 312: Table 4. Please fix the format of this table.

R. Done

Round 3

Reviewer 2 Report

All my questions have been successfully answered by authors and the manuscript, in my opinion, is publishable in its current form.

Congratulations to authors and happy new year.